# The COVID-19 Pandemic and Patient Safety Culture: A Cross-Sectional Study among Community Pharmacies in Jordan

**DOI:** 10.3390/healthcare10081434

**Published:** 2022-07-30

**Authors:** Mohammad Abu Assab, Deema Jaber, Haneen Basheer, Hanadi Abu Assab, Haya Al-Atram

**Affiliations:** 1Department of Clinical Pharmacy, Faculty of Pharmacy, Zarqa University, Zarqa 13110, Jordan; dsuleiman@zu.edu.jo (D.J.); hbasheer@zu.edu.jo (H.B.); 20209318@zu.edu.jo (H.A.A.); 2Al-Jalodi Pharmacy, Al-Mafraq 25110, Jordan; hayaheyasat96@yahoo.com

**Keywords:** COVID-19, patient safety culture, community pharmacy, healthcare quality, pharmaceutical care, Jordan

## Abstract

The COVID-19 pandemic has dramatically imposed stressful conditions that may impact the ability of healthcare staff to provide safe and effective care. Research on patient safety culture among community pharmacies during the pandemic is limited. This study aimed to assess the patient safety culture among community pharmacies in Jordan during the COVID-19 pandemic. Pharmacists and pharmacy assistants from 450 community pharmacies were approached through online means, with 378 answering the questionnaire written in Arabic that had been adapted from the Community Pharmacy Survey on Patient Safety Culture (PSOPSC). This study showed that various patient safety standards were addressed to a high degree during the COVID-19 pandemic, as represented by the high positive response rate (PRR) measures that were mainly observed in the dimensions “Teamwork” (90.1%), “Patient Counseling” (85.2%), and “Staff Training and Skills” (82.7%). Furthermore, significantly higher PRR scores for the “Teamwork”, “Staffing, Work Pressure, and Pace”, “Response to Mistakes”, “Organizational Learning—Continuous Improvement”, and “Overall Perceptions of Patient Safety” dimensions were observed among participants who worked in independent pharmacies than those who worked in chain pharmacies. Despite an overall positive patient safety culture in the current context of community pharmacies in Jordan during the COVID-19 pandemic, pitfalls were observed in the “Staffing, Work Pressure, and Pace” dimension.

## 1. Introduction

COVID-19 has significantly strained healthcare systems, resulting in far-reaching consequences for how healthcare is delivered. During the pandemic, rapid changes in various healthcare delivery models were observed, including increased workloads, the redeployment of staff to unfamiliar clinical environments, the cancellation of routine services, and the requirement to treat patients suffering from a novel disease about which little was known. Working in these stressful conditions had the potential to impact the ability of healthcare staff to provide safe and effective care [1,2,3].

The provision of quality healthcare encompasses patient safety, which can be defined as “the freedom from accidental injuries during medical care, activities to avoid, prevent, or correct adverse outcomes, which result from the delivery of healthcare” as a strategic matter [4]. Therefore, adopting a safety culture in healthcare organizations is one of the main strategies used to achieve patient safety objectives and to improve the quality of healthcare services. An organization’s safety culture is the product of individual and group values, attitudes, perceptions, competencies, and patterns of behavior that determine the commitment to and the style and proficiency of an organization’s health and safety management [5]. This definition applies to all healthcare organizations and aims to reduce, prevent, or avoid dreadful mistakes, adverse events, and mortality [6]. Many patient safety subcultures have been identified, including leadership, teamwork, evidence-based, communication, learning, and patient-centered approaches [7].

According to the Institute of Medicine, the movement towards a safer health system is changing the culture from one of blaming individuals for errors to one that does not treat errors as personal faults and that instead treats them as opportunities to prevent harm and to improve the system [8]. Thus, developing a safety culture has become a central aspect in the efforts of many healthcare organizations to improve the quality of patient safety, treatment, the care process, and the system as a whole. Additionally, several international health bodies, such as the World Health Organization (WHO) and the Joint Commission International (JCI), have advocated for the measurement of safety culture within healthcare organizations as an effective strategy to improve sustainable safety. Consequently, multiple assessment tools have been articulated and used to evaluate the extent of safety culture in healthcare organizations, particularly in hospital settings [9,10,11,12].

Multiple studies have also been conducted to assess patient safety culture in the hospitals and primary healthcare settings of some Arab countries [13,14,15,16]. In China, a study explored the attitudes of hospital pharmacy workers using a pharmacy survey; the results showed a positive attitude toward patient safety culture in the participants’ organizations [17]. Assessing patient safety culture in healthcare organizations during the ongoing COVID-19 pandemic has mainly been addressed in hospital settings, where multiple studies have been conducted worldwide [1,2,3].

Community pharmacies are crucial healthcare organizations because of their high patient accessibility, their direct interaction with the public, and their wide range of services. Locally, community pharmacies are plentiful, highly accessible, and vastly distributed. They offer prescription and over-the-counter medications, cosmetics, skincare products, and medical equipment [18]. They deal almost with 8000 pharmaceuticals, 3000 of which are over-the-counter drugs. There is no practice framework for providing patient-centered pharmaceutical care, and the primary duty of community pharmacists is to supply products. Patients visiting independent or chain pharmacies receive patient counseling neither regularly nor systematically [19].

Additionally, Jordanian community pharmacies handle the prescriptions brought by patients and do not use an electronic system that links physician orders directly to the pharmacy. A study in Jordan showed that the total number of medication dispensing errors averaged 24.6% for dispensed medications; this included pharmacist counseling errors (11.5%) and prescription-related errors (13.1%). Most of these errors were made by pharmacists (79.6%), followed by pharmacy assistants (12.4%) [20].

Therefore, community pharmacies have continuously attempted to improve the quality of health and pharmaceutical care services provided to patients, focusing on optimizing patient safety and decreasing unintended mistakes during healthcare delivery. Before the COVID-19 pandemic, several studies assessed patient safety culture in community pharmacy settings both internationally and regionally [17,21,22,23,24,25,26,27,28]. Previous research has identified organizational breakdown, insufficient staffing, increased production pressures, and provider fatigue as factors contributing to poor patient safety [29,30]. All of these factors could have existed among community pharmacies during the pandemic. As a result, there is a substantial need to assess patient safety culture during the COVID-19 pandemic. 

The objective of this study was to assess the current patient safety culture dimensions and the practices adopted by pharmacists and pharmacy assistants in community pharmacies in Jordan during the ongoing COVID-19 pandemic.

## 2. Materials and Methods

### 2.1. Study Design and Settings

The research was a descriptive, analytical cross-sectional study that used a pre-tested, validated questionnaire that was electronically distributed to a sample of community pharmacies in Jordan during the COVID-19 pandemic. Data were collected over three months (from April to July 2021).

### 2.2. Study Instrument

The study questionnaire was adapted from the Community Pharmacy Survey on Patient Safety Culture (PSOPSC), which was originally developed by the Agency for Healthcare Research and Quality (AHRQ) [31]. The PSOPSC is a comprehensive tool with previously proven validity and reliability in its dimensions and items and was explicitly developed for community pharmacy settings. The questionnaire was translated into Arabic to support the participants in their responses. Initially, it was translated into Arabic and revised for formal language wording. According to the revision feedback, responses such as “Does Not Apply/Do not Know” were omitted, as they were expected to confuse respondents. Subsequently, a pilot study was conducted to ensure that the questions were straightforward and that they reflected the study’s objectives. Sixty community pharmacies were randomly selected, and the questionnaire and the study objectives were sent to them electronically, inviting them to participate. Electronic responses to the pilot questionnaire were collected over two weeks. Cronbach’s α was used to measure the questionnaire’s overall internal consistency and reliability [32].

In the qualitative content validity assessment, ten experts were asked to comment on the items regarding their grammar, choice of vocabulary, placement, and scoring. Furthermore, a group of clinical pharmacists, a statistician, and a sociologist individually evaluated the prepared questionnaire to ensure face validity. The evaluators assessed the difficulty, generality, and ambiguity of the items. Subsequently, the pilot study was conducted, and the updated, refined questionnaire was distributed electronically through social media, where pharmacists and pharmacy assistants (the targeted respondents) were invited to participate.

### 2.3. Sample Size Calculation and Sampling Strategy

According to the Jordan Pharmacists Association, there are currently around (3500) community pharmacies and approximately 8000 community pharmacists and pharmacy assistants. The minimum sample size required for a 95% confidence level and a 5% margin of error (significance α = 0.05) with a 50% response distribution was found to be 367 [33]. The questionnaire was electronically distributed to 450 community pharmacies via their dedicated WhatsApp and Facebook groups.

### 2.4. Inclusion and Exclusion Criteria

The study respondents were pharmacists and pharmacy assistants working in community pharmacies in Jordan. All other pharmacy staff members were excluded from this study.

### 2.5. Questionnaire Measures

The questionnaire consisted of two parts: the first part was designed to obtain the socio-demographic characteristics of the respondents, and the second was dedicated to investigating the dimensions of patient safety culture. The second part included 41 items distributed among six sections. Sections A (10 items; A1–A10), B (16 items; B1–B16), and C (10 items; C1–C10) were dedicated to measuring the 11 composites of patient safety culture. Appendix A shows the detailed patient safety culture composites with their definitions.

Sections D (3 items; D1–D3), E (1 item), and F (1 item) were assigned to determine the frequency mistake documentation, the overall patient safety rating, and open-ended comments regarding issues affecting patient safety, respectively.

### 2.6. Study Endpoints

The primary endpoints were to assess the current status of adopting patient safety culture dimensions and composites represented by the positive response rate (PRR), the extent of mistake documentation, and the overall patient safety rating among community pharmacies in Jordan during the COVID-19 pandemic.

The secondary endpoints were outcomes related to the correlations between the participant characteristics and patient safety culture dimensions and to identify areas where patient safety culture could be improved.

### 2.7. Data Analysis

The collected data were transported to the Statistical Package for Social Science (SPSS) software, version 23, for analysis. The participant demographics were presented using descriptive statistics (frequency/percentage). The composite frequencies of the eleven patient safety culture dimensions and the positive response rates (PRRs) for the survey questions were calculated as per the user guide for PSOPSC published by the AHRQ [31]. Most of the questionnaire items (Sections A, B, and C; 36 items) used a five-point Likert agreement response scale (strongly disagree to strongly agree). A five-point Likert frequency response scale (never to always) was used for the items in section D. For positively worded items (positive practice statements), the PRR of each item was calculated by combining the response frequencies of the two highest Likert scale response categories (e.g., strongly agree and agree) and divided by the total number of respondents. In contrast, negatively worded items (negative practice statements that respondents are expected to disagree with if they adopt a positive safety culture in their pharmacies) were reverse coded (R) during PRR calculation. Explicitly, PRR was performed for these questions by adding the frequencies of the lowest two response categories on the Likert scale (e.g., strongly disagree and disagree) and dividing them by the total number of respondents. Respondents rated the “Overall Rating on Patient Safety” in their community pharmacy (section E) as “poor,” “bad,” “good,” “very good,” or “excellent.” Correlations between socio-demographic characteristics and PRR were assessed using the chi-square and Fisher’s exact tests, with a significance level of *p* < 0.05.

## 3. Results

A total of 450 community pharmacies from ten governorates were approached to participate in the study, 378 of which answered the questionnaire, resulting in a response rate of (84.0%). 

### 3.1. Socio-Demographic Characteristics of the Participants

The socio-demographic characteristics of the participants are presented in Table 1. Most of the participants were female (*n* = 253, 66.9%), had a BSc degree in pharmacy (*n* = 296, 78.3%), worked in independent pharmacies (*n* = 292, 77.2%), and had six or more years of working experience in community pharmacies (*n* = 230, 60.8%).

### 3.2. Reliability of the Questionnaire

The questionnaire’s consistency and reliability were measured using Cronbach’s α test, which determined that the overall internal consistency of the study was excellent (Cronbach’s α = 0.910).

### 3.3. Safety Culture Assessment across Community Pharmacies in Jordan during the COVID-19 Pandemic

Table 2 shows the participants’ perceptions of patient safety culture standards and dimensions in their pharmacies during the COVID-19 pandemic. The highest PRR score was generally observed in the “Working in this pharmacy” standards (84.6%), followed by the “Patient safety and response to mistakes” and “Communication and work pace” standards, which scored 76.4% and 70.9%, respectively. For the patient safety culture dimensions, the “Teamwork” dimension (90.1%) scored the highest PRR, followed by the “Patient Counseling” and “Staff Training and Skills” domains, which scored 85.2% and 82.7%, respectively. The “Staffing, Work Pressure, and Pace” dimension scored the lowest PRR among all of the patient safety culture dimensions (46.6%). The overall PRR of all 36 items was (77.3%). The PRR for a negative practice item, B9: “We feel rushed when processing prescriptions,” was 26.2%. Thus, only 26.2% of respondents disagreed with the negative practice described in B9; the participants do not feel rushed when processing prescriptions in their pharmacies; however, the remaining respondents (73.8%) feel rushed when processing prescriptions in their pharmacies.

The “overall perception of patient safety” dimension, which was assessed through three items, reported a PRR of (78.5%), with a strong focus on patient safety (84.4%) and a good ability to present mistakes (84.1%).

The participants were asked about the frequency of mistakes reported in the pharmacy, and their responses are shown in Table 3. Of note, mistakes were not regularly documented during the COVID-19 pandemic crises, regardless of their impact.

Additionally, the analysis of the results revealed that the participants who worked in independent pharmacies scored a significantly higher average rate for the “Teamwork” (*p* = 0.037) dimension, the “Staffing, Work Pressure, and Pace” dimension (*p* = 0.018), the “Response to Mistakes” dimension (*p* = 0.001), the “Organizational Learning—Continuous Improvement” dimension (*p* = 0.001), and the “Overall Perceptions of Patient Safety” dimension (*p* = 0.006) compared to those who worked in chain pharmacies. Appendix A portrays the detailed PRRs for patient safety culture items and dimensions according to pharmacy ownership type.

## 4. Discussion

This study is the first to be conducted during the ongoing COVID-19 pandemic that investigates the practices and perspectives of community pharmacists and pharmacy assistants regarding the patient safety culture in Jordan. Community pharmacies are significantly different from pharmacies in hospitals in terms of the work context, workflow system, and staff responsibilities. Therefore, to ensure the best safety culture practices among community pharmacies in Jordan during the COVID-19 pandemic, an assessment of the status of the current safety culture is needed.

Compared to pre-COVID-19 studies, the overall PRR in the present study for the 36 items of the patient safety culture questionnaire was (77.3%), which is comparable to a study conducted in Abu Dhabi (74.7%) but less than the PRR obtained in a Kuwaiti study (83.3%) and higher than that obtained in both Malaysian research (67.0%) and in a Saudi study (60.2%) [21,22,23,26]. Thus, we can conclude the existence of a positive patient safety culture among community pharmacies in Jordan during the COVID-19 pandemic, as represented by the high PRR measures of the patient safety composites (scores ranged from 46.6% to 90.1%) and the overall PRR for the 36 items of the patient safety culture questionnaire.

Regarding the PRR across the patient safety culture dimensions during the COVID-19 pandemic, the “Teamwork” dimension scored the highest PRR (90.1%). This finding emphasizes the importance of teamwork during the pandemic as being critical to the patient safety practices that produce particular efficiencies and communication patterns, thereby increasing the capacity of pharmacies to deal with pandemic-related work surges in the community pharmacy context [34,35].

Despite there being less direct contact and the changes in communication between patients and pharmacists due to the guidelines implemented by the COVID-19 pandemic health authorities, “Patient Counseling” obtained the second-highest PRR (85.2%) in our study [36]. According to the Jordanian Medication and Pharmacy Law, patient counseling is an obligatory responsibility of community pharmacies. Furthermore, many studies have shown that community pharmacists’ engagement with patients positively enhances therapeutic outcomes [37]. To compensate for the lack of counseling in the pharmacy, community pharmacists provided patients with online informational materials and counseled patients using mobiles and telephones [36].

The “Staff Training and Skills” dimension scored 82.7% and came third in the PRR across the patient safety culture dimensions. The issue reflects the positive impact of proper orientation, training, and skills among community pharmacy staff members to perform their roles properly [38]. Early COVID-19 experience suggests the central importance of computer skills and comfort with the rapidly evolving technologies for virtual care/clinical consultation and digital dispensing to support practice resilience among community pharmacists. The effective use of these technologies by trained and confident pharmacists is critical to managing the pandemic-linked workload surges [39].

This study revealed that the lowest PRR score among all of the patient safety culture dimensions (46.6%) was reported for the “Staffing, Work Pressure, and Pace” dimension. Within this dimension, the following items scored the lowest PRRs of 23.5%, 26.2%, and 59.8% for frequent interruptions or distractions in the pharmacy, feeling rushed when processing prescriptions, and inadequate breaks during shifts, respectively. It has been shown that an unfavorable working environment leads to increased dispensing errors [40]. Worldwide, the workload of community pharmacists has increased dramatically during the COVID-19 pandemic, with standard workflow and work-related activities being disrupted. Furthermore, employment changes are associated with anxiety, resistance, insecurity, loss of control, and general inconvenience [41]. Managing crowdedness in community pharmacies, which requires people to take turns and wait outside until the crowding dissipates; deep cleaning protocols; managing deliveries; and stocking shelves were described as being highly stressful for all of the staff members during the early weeks of the pandemic [39]. Training and education could be helpful strategies for increasing employee adaptability, but these were not possible during the pandemic. A study showed that despite workflow changes being incorporated to simplify and coordinate the work of pharmacists during the pandemic, most pharmacists found them to be more complex, and many shared negative thoughts and experiences with the new workflow system [41]. 

Of note, our results showed that the scores for the communication practices for patient safety culture were lower than those of other standards. In particular, the “Communication about Prescriptions across Shifts” and the “Communication openness” dimensions showed relatively lower scores of 72.4% and 78.8%, respectively. The use of face masks and the implementation of social distancing guidelines during the pandemic make it more challenging to understand and communicate effectively with patients [42]. Therefore, there is an urgent need to improve the communication aspect of patient safety culture to minimize drug-related problems. Furthermore, continuous communication about dispensing errors is a pillar of adopting a patient safety culture in healthcare organizations. An analysis of dispensing errors to determine their cause and to detect them effectively is necessary to prevent future errors [43,44,45]. During the COVID-19 pandemic, community pharmacies and pharmacists who felt comfortable and confident managing electronic communication reported having more control over their workflow and being able to triage and queue patients more effectively based on priority and need [39].

Another patient safety culture gap is error documentation within community pharmacies and other healthcare organizations. This assessment revealed that regardless of the risk associated with the error, the documentation level during the pandemic among the community pharmacies that participated was considerably low. Recently, the Jordanian government articulated and released the “Law of Medical and Health Responsibility,” in which dispensing errors are considered the liability of public health practitioners, similar to in other countries.

An analysis of the current study results obtained during the COVID-19 pandemic showed that participants who worked in independent pharmacies scored significantly higher average rates for the “Teamwork” (*p* = 0.037) dimension, the “Staffing, Work Pressure, and Pace” dimension (*p* = 0.018), the “Response to Mistakes” dimension (*p* = 0.001), the “Organizational Learning—Continuous Improvement” dimension (*p* = 0.001), and the “Overall Perceptions of Patient Safety” (*p* = 0.006) than those who worked in chain pharmacies. We can assume that workload, working conditions, and environment differ significantly between independent and chain pharmacies. A study showed that the surge in demand for specific pharmaceutical products during the COVID-19 pandemic increased the workload among community pharmacies, disproportionately impacting pharmacists in larger chains. Additionally, client behavior was less pleasant for pharmacists working in these chains [41]. Knowing that chain pharmacies constitute approximately 10.0% of Jordan’s total number of community pharmacies, a comprehensive investigation of various aspects of pharmaceutical care delivery among independent and chain community pharmacies comprises an area of future research.

### 4.1. Study Strengths and Limitations

This research is the first study concerning patient safety culture within community pharmacies in Jordan conducted during the COVID-19 pandemic. The study included community pharmacists and pharmacy assistants among individual and chain pharmacies. The electronically distributed questionnaire was utilized as a data collection tool due to the governmental pandemic-related restrictions and measures. The participants in the study were homogenous in terms of their nationality, as only Jordanian pharmacists are allowed to own a pharmacy, resulting in most community pharmacists and assistants being Jordanian. Thus, response bias due to a fear of negative responses having a negative impact on a pharmacy’s reputation or participants’ job security is unlikely to occur. The results of this study cannot be generalized to pharmacists or pharmacy assistants working in healthcare institutions other than community pharmacies due to significant differences in the work environment, workflow system, and responsibilities.

### 4.2. Implications for Future Research

This research comprises fertile ground for conducting a follow-up study after the COVID-19 pandemic has ended to examine change trends in the patient safety culture over time and for conducting regional and international comparisons. Additionally, assessing the impact of patient characteristics on patient safety culture, evaluating the impact of the patient–pharmacist relationship on patient safety culture, evaluating the cultural impact of patient safety culture initiatives and interventions, and investigating various aspects of pharmaceutical care delivery among independent and chain community pharmacies represent areas of interest for future research.

### 4.3. Policy Implications

The findings of this study drive the attention of policymakers, community pharmacy owners, and decision-makers to embrace effective policies, including flexi-time, the adoption of flexible workflow systems improvements in the working environment, and the presence of an adequate number of staff members as valuable strategies to support serving patients correctly. The lack of documentation of mistakes within community pharmacies is another revealed patient safety culture gap that requires healthcare policymakers to disseminate guidelines that address the identification and reporting of errors. Furthermore, community pharmacies in Jordan should articulate plans, develop policies, and implement strategies to improve areas of patient safety, particularly regarding communication about prescriptions across shifts, responses towards mistakes, and communication openness. Long-term commitment to bridging the identified gaps, particularly under the stressful work conditions caused by COVID-19, is vital for health policymakers and community pharmacy owners to improve the quality of healthcare during the delivery of pharmaceutical care.

## 5. Conclusions

This study acknowledges the perspectives of community pharmacists and their assistants toward patient safety culture. It revealed the existence of an overall positive patient safety culture in the context of community pharmacies in Jordan during the COVID-19 pandemic. Despite that, pitfalls were primarily observed in the “Staffing, Work Pressure, and Pace” dimension. Bridging the identified gaps is crucial to improving the quality of healthcare during the delivery of pharmaceutical care.

## Figures and Tables

**Table 1 healthcare-10-01434-t001:** Socio-demographic details of respondents (N = 378).

Parameter		N (%)
Age (in years)	Less than 25	54 (14.3%)
	25 to less than 35	127 (33.6%)
	35 to less than 45	80 (21.2%)
	45 or older	117 (31.0%)
Gender	Female	253 (66.9%)
	Male	125 (33.1%)
Education level	Bachelor of Pharmacy	296 (78.3%)
	Bachelor of Pharm D	11 (2.9%)
	Postgraduate degree	29 (7.7%)
	Pharmacy diploma	42 (11.1%)
Pharmacy location (Governorates)	Middle area, including the capital, Amman	305 (80.7%)
	North area	41 (10.8%)
	South area	32 (8.5%)
Pharmacy ownership type	Independent (individual)	292 (77.2%)
	Chain pharmacy	86 (22.8%)
The position in the pharmacy (Job title)	Staff pharmacist	82 (21.7%)
	Responsible pharmacist	119 (31.5%)
	Pharmacy assistant	40 (10.6%)
	Owner pharmacist	121 (32.0%)
	Other	16 (4.2%)
Years of practical experience in community pharmacies	Less than 3 years	99 (26.2%)
	3 years to less than 6 years	49 (13.0%)
	6 years or more	230 (60.8%)
Weekly working hours	Less than 16 h per week	41 (10.8%)
	17 to 31 h per week	54 (14.3%)
	32 to 40 h per week	110 (29.1%)

**Table 2 healthcare-10-01434-t002:** Positive response rates (PRRs) of dimensions and individual items across participating community pharmacies during the COVID-19 pandemic (N = 378).

Total Positive Response Rates (PRRs) of all Sections (77.3%)	PRR	
	N	%
**SECTION A: Working in this Pharmacy (section positivity)**		84.6
**1. Physical Space and Environment (dimension positivity):**		81.7
A1 This pharmacy is well organized	333	88.1
A5 This pharmacy is free of clutter	285	75.4
A7 The physical layout of this pharmacy supports good workflow	308	81.5
**2. Teamwork (dimension positivity):**		90.1
A2 Staff treat each other with respect	355	93.9
A4 Staff in this pharmacy clearly understand their roles and responsibilities	335	88.6
A9 Staff work together as an effective team	332	87.8
**3. Staff Training and Skills (dimension positivity):**		82.7
A3 Technicians in this pharmacy receive the training they need to do their jobs	301	79.6
A6 Staff in this pharmacy have the skills they need to do their jobs well	323	85.4
A8 Staff who are new to this pharmacy receive adequate orientation	320	84.7
A10 Staff get enough training from this pharmacy	307	81.2
**SECTION B: Communication and Work Pace (section positivity)**		70.9
**4. Communication Openness (dimension positivity):**		78.8
B1 Staff ideas and suggestions are valued in this pharmacy	289	76.5
B5 Staff feel comfortable asking questions when they are unsure about something	309	81.7
B10 It is easy for staff to speak up to their supervisor/manager about patient safety concerns in this pharmacy	295	78.0
**5. Patient Counseling (dimension positivity):**		85.2
B2 We encourage patients to talk to pharmacists about their medications	329	87.0
B7 Our pharmacists spend enough time talking to patients about how to use their medications	314	83.1
B11 Our pharmacists tell patients important information about their new prescriptions	323	85.4
**6. Staffing, Work Pressure, and Pace (dimension positivity):**		46.6
B3 Staff take adequate breaks during their shifts	226	59.8
B9 We feel rushed when processing prescriptions (R)	99	26.2
B12 We have enough staff to handle the workload	290	76.7
B16 Interruptions/distractions in this pharmacy (from phone calls, faxes, customers, etc.) make it difficult for staff to work accurately (R)	89	23.5
**7. Communication About Prescriptions Across Shifts (dimension positivity):**		72.4
B4 We have clear expectations about exchanging important prescription information across shifts	272	72.0
B6 We have standard procedures for communicating prescription information across shifts	255	67.5
B14 The status of problematic prescriptions is well communicated across shifts	294	77.8
**8. Communication About Mistakes (dimension positivity):**		79.5
B8 Staff in this pharmacy discuss mistakes	308	81.5
B13 When patient safety issues occur in this pharmacy, staff discuss them	292	77.2
B15 In this pharmacy, we talk about ways to prevent mistakes from happening again	301	79.6
**SECTION C: Patient Safety and Response to Mistakes (section positivity)**		76.4
**9. Response to Mistakes (dimension positivity):**		72.5
C1 Staff are treated fairly when they make mistakes	272	72.0
C4 This pharmacy helps staff learn from their mistakes rather than punishing them	298	78.8
C7 We look at staff actions and the way we do things to understand why mistakes happen in this pharmacy	306	81.0
C8 Staff feel like their mistakes are held against them (R)	220	58.2
**10. Organizational Learning—Continuous Improvement (dimension positivity):**		79.6
C2 When a mistake happens, we try to figure out what problems in the work process led to the mistake	315	83.3
C5 When the same mistake keeps happening, we change the way we do things	310	82.0
C10 Mistakes have led to positive changes in this pharmacy	278	73.5
**11. Overall Perceptions of Patient Safety (dimension positivity):**		78.5
C3 This pharmacy places more emphasis on sales than on patient safety (R)	253	66.9
C6 This pharmacy is good at preventing mistakes	318	84.1
C9 The way we do things in this pharmacy reflects a strong focus on patient safety	319	84.4

(R): Negatively worded items were reversed coded to calculate the PRR of the item.

**Table 3 healthcare-10-01434-t003:** Frequency of mistakes reported by community pharmacists (N = 378).

Documenting Mistakes	* PRR		
	Never/Rarely N (%)	Sometimes N (%)	(Most of the Time/Always) N (%)
D1 When a mistake reaches the patient and could cause harm but does not, how often is it documented?	112 (29.6%)	100 (26.5%)	166 (43.9%)
D2 When a mistake reaches the patient but has no potential to harm the patient, how often is it documented?	107 (28.3%)	127 (33.6%)	144 (38.1%)
D3 When a mistake that could have harmed the patient is corrected before the medication leaves the pharmacy, how often is it documented?	72 (19.0%)	100 (26.5%)	206 (54.5%)

* PRR: positive response rate.

## Data Availability

The data presented in this study are available on request from the corresponding author.

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
