# Peer review of "The COVID-19 Pandemic and Patient Safety Culture: A Cross-Sectional Study among Community Pharmacies in Jordan"

_healthcare, 2022, doi:10.3390/healthcare10081434_

Round 1
Reviewer 1 Report
Need more work for better presentation of your findings
Introduction
Referencing: example, delivery of healthcare" [4] à reference should be at the end of the sentence. In your manuscript, there are many citations in the middle of sentence which is not correct. Please fix this and all other references throughout your manuscript
“This study came to assess the” please change to “the objective of this study”
Paragraph that start with Abdul-Fatouh et al.. I think it is not necessary and making the introduction very long. I recommend you delete from introduction. You may put some of these details in the discussion section if needed and relevant
Method:
You need to put subsections. For example, 2.1 Study design, 2.2 … etc
Add reference here “was adapted from the Community 90 Pharmacy Survey on Patient Safety Culture (PSOPSC), initially developed by the Agency 91 for Healthcare Research and Quality (AHRQ).”
Add reference for Cronbach's α
Missing period “to be a (367) [29] Th”
What are your primary and secondary end points? You need to add section in the method section.
The method is long, if there is anything can be moved to supplement it would be good idea.
Results:
“average rate for the "Teamwork" (p = 208 0.037) dimension, the "Staffing, Work Pressure, and Pace" dimension (p = 0.018), the 209 "Response to Mistakes" dimension (p = 0.001), the "Organizational Learn-“ here font is different make sure to fix and be consistent
So many tables. Are all of these needed in the manuscript? Can you move some of them to the supplementary materials? This can confuse the reader and loose interest
Discussion:
These 2 paragraphs should be merged into one
“Of note, our results showed that the communication practices” and “Therefore, there is an urgent need to improve the commu”
4.1 strength and limitation s
Should be one paragraph no need to be 3
Discussion. Overall very long, try to cut unnecessary details
Reviewer 2 Report
Patient safety culture might be self-defining, but I would like to see a definition and a discussion of what is included in your use of the phrase. What are some of the specifics included in the relationship between the patient and the pharmacist?
Line 95. Not clear what designed in Arabic means. Is it just the language or are there other differences? I also don't understand the sentence in line 96. what is the formal language?
It would be useful to know how the pharmacies work. Are prescriptions sent from clinicians to the pharmacy? Do patients bring prescriptions? are some drugs sold over the counter? What is the nature of the errors that occur? Is it the wrong patient? Is it the wrong dose? is it the wrong drug? To understand your study, I think it is important to know more details about the scene you are evaluating. I also would like to know something about the magnitude of the errors. Do the patients spot errors when they occur? Would patient characteristics impact the findings of this study?
Line 194: not clear to me what R means by reversed coded. For example, for B9, does that mean that 26.2% feel rushed? Please give a word example to make the understanding very clear. Line 196: What impact does the lack of documentation have on your study?
Your comparisons were with mostly Arab countries as appropriate. would it make sense to try this experiment in non-Arab counties such as UK, France, or the United States?
Was anyone ever fired because of a mistake?
What happens after the study is completed and published? Will behavior changes occur? Will you do a follow-up study after Covid calms?
Round 2
Reviewer 1 Report
None